# Multiple spawning run contingents and population consequences in migratory striped bass *Morone saxatilis*

David H. Secor[1]*, Michael H. P. O'Brien[1], Benjamin I. Gahagan[2], Dewayne A. Fox[3], Amanda L. Higgs[4], Jessica E. Best[4]

1 University of Maryland Center for Environmental Science, Chesapeake Biological Laboratory, Solomons, Maryland, United States of America, 2 Massachusetts Division of Marine Fisheries, Gloucester, Massachusetts, United States of America, 3 Delaware State University, College of Agriculture, Science, and Technology, Dover, Delaware, United States of America, 4 Division of Marine Resources, New York State Department of Environmental Conservation, Department of Natural Resources, Cornell University, New Paltz, New York, United States of America

* secor@umces.edu

**Data Availability Statement:** All data are available in, O'Brien, Michael et al. (2020), Multiple spawning run behavior and population consequences in migratory striped bass Morone

## Abstract

Multiple spawning run contingents within the same population can experience varying demographic fates that stabilize populations through the portfolio effect. Multiple spawning run contingents (aka run timing groups) are reported here for the first time for striped bass, an economically important coastal species, which is well known for plastic estuarine and shelf migration behaviors. Adult Hudson River Estuary striped bass (n = 66) were tagged and tracked with acoustic transmitters from two known spawning reaches separated by 90 km. Biotelemetry recaptures for two years demonstrated that each river reach was associated with separate contingents. Time series of individual spawning phenologies were examined via nonparametric dynamic time warping and revealed two dominant time series centroids, each associated with a separate spawning reach. The lower spawning reach contingent occurred earlier than the higher reach contingent in 2017 but not in 2018. The majority (89%) of returning adults in 2018 showed the same contingent behaviors exhibited in 2017. Spawning contingents may have been cued differently by temperatures, where warming lagged 1-week at the higher reach in comparison to the lower reach. The two contingents exhibited similar Atlantic shelf migration patterns with strong summer fidelity to Massachusetts Bay and winter migrations to the southern US Mid-Atlantic Bight. Still, in 2017, differing times of departure into nearby shelf waters likely caused the early lower reach contingent to experience substantially higher mortality than the later upper reach contingent. Anecdotal evidence suggests that higher fishing effort is exerted on the early-departing individuals as they first enter shelf fisheries. Thus, as in salmon, multiple spawning units can lead to differential demographic outcomes, potentially stabilizing overall population dynamics.

saxatilis, Dryad, Dataset, https://doi.org/10.5061/dryad.6hdr7sqxt.

**Funding:** DHS and ALH received Hudson River Foundation https://www.hudsonriver.org/ (Grant 011/15A) support for this study. The funders had no role in study desing, data collection and analysis, decition to publish, or preparation of the manuscript.

**Competing interests:** The authors have declared that no competing interests exist.

# Introduction

Spawning migrations are periods of heightened vulnerability, the outcome of which drives population dynamics [1–4]. Extreme gonad provisioning and swimming expenditures cause death in some fishes. Further, the restricted and predictable migration routes of spawners lead to increased exposure to environmental degradation and catastrophe, and, in some species, severe vulnerability to predation and fishing exploitation. A key buffer against these sources of mortality is multiplicity in spawning runs [5]. Discrete spawning runs and units within runs (referenced here as run contingents and in the salmon literature as run timing groups [1]) with differential timing, routes, and endpoints will each experience different mortality regimes that are dynamic across generations. Through the stabilizing feature described initially as response diversity [6,7] and then the portfolio effect [7,8], multiple demographic outcomes among spawning runs can stabilize population dynamics.

The best studied spawning runs, those of salmons, shads, and river herrings, are overt. Such fishes push up from coastal waters into shallow and narrow confines of non-tidal fresh-water ecosystems, where their abundance and schooling behaviors are often on full display [9–11]. Indeed, these oft-depicted spawning runs represent the quintessential fish migration in public understanding [12–15]. The spawning migrations that occur in coastal waters are more difficult to observe but here, too, multiple spawning run behaviors can occur in the same population. For instance, both spring and fall spawning occur in populations of Atlantic herring [16,17], Atlantic cod [18], and Atlantic sturgeon [19]. Still, more-nuanced diversity in spawning phenologies, such as those that occur within the same run, remain undescribed for coastal species.

This study documents multiple spawning-run contingents for striped bass *Morone saxatilis*, a ubiquitous predator in NW Atlantic shelf waters, which supports important commercial and recreational fisheries. Striped bass are moderately long-lived, late maturing, and on average spawn each year of their adult life, although skipped spawning does occur [20]. Spawning migrations, some exceeding 1000 km, occur from shelf to estuarine waters each spring [20,21]. Fisheries target and intercept spawners as they arrive and depart spawning reaches [22,23]. Typically, spawning is concentrated just above the salt front [24,25], yet in larger estuaries multiple areas of concentrated spawning occur [26]. We hypothesized that in one such estuary, the Hudson River Estuary, at least two spring spawning run contingents occur, each associated with known centers of egg production (Fig 1).

To be of ecological consequence, each spawning run contingent should exhibit characteristic migration behaviors, repeat these behaviors across years, and encounter varying mortality regimes. Run contingents are defined as a cycle of directed up-estuary and down-estuary migration behaviors occurring within the Hudson River Estuary, and are classified by both phenology (timing) and spawning reach (up-estuary extent). We leave alone the question of evolutionary and conservation consequences associated with genetic differentiation between contingents [28], and rather examine ecological consequences of their behaviors following their departure. [29,30]. Through biotelemetry and time series classification using dynamic time warping, we asked; (1) Are characteristic spatiotemporal migration behaviors repeated within successive spring spawning runs of the tagged individuals? (2) Do the same individuals undertake the same contingent behavior between years? (3) Do spawning run contingents undertake the same migration behaviors during non-spawning periods? and, (4) Do mortality rates vary between spawning run contingents? Within the limits of two years of spawning phenologies, we also evaluated the influence of temperature, which has been identified as a dominant spawning cue for striped bass [31–33].

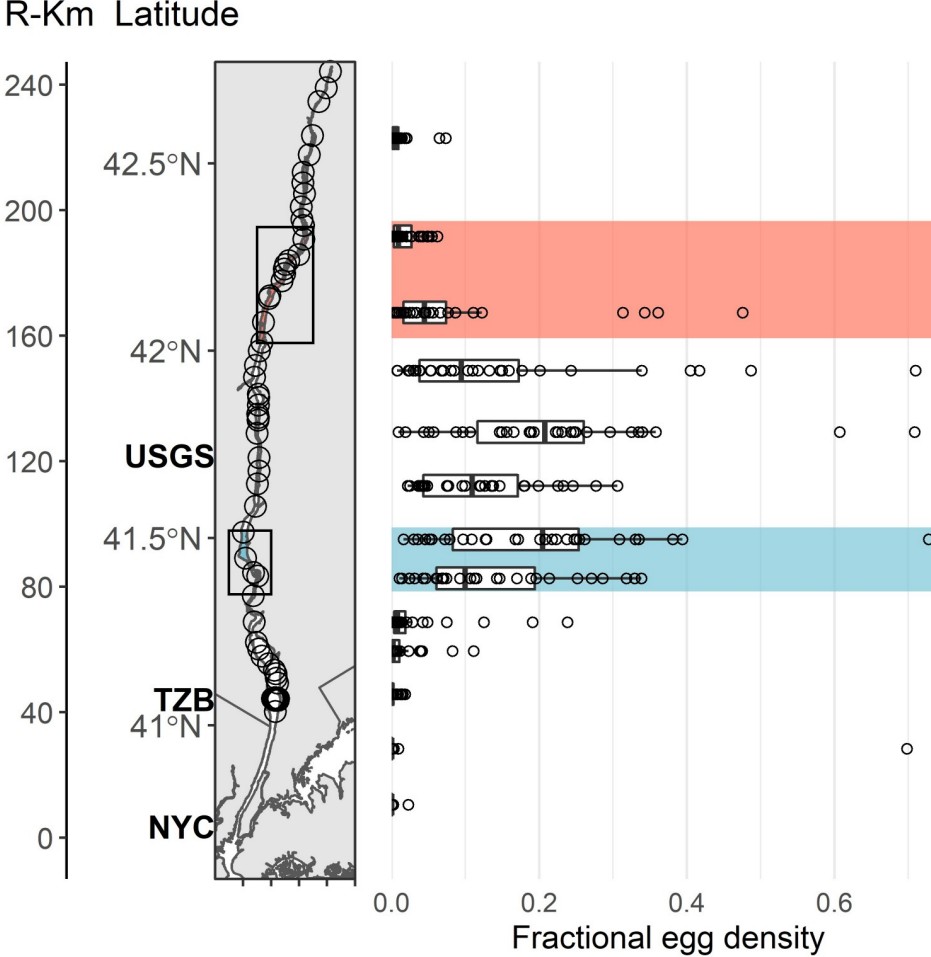

**Fig 1. Map of the Hudson River, NY, USA and distribution of striped bass eggs.** Data digitized from the Hudson River Long River Ichthyoplankton Survey, 1976–2012 (references to all annual reports containing the data are listed in [27]). Y-axis shows river km (R-Km) and latitude. Circles on the map display deployed telemetry receivers, and the location of New York City (NYC), the Tappan Zee Bridge (TZB), and the Poughkeepsie USGS water monitoring station (USGS) are shown. Points show yearly values of egg density and box plots represent interannual variation. Highlighted areas, upper reach (red) and lower reach (blue) represent the regions of spawner abundance monitored by NY state scientists.

## Materials and methods

### Study species and system

Natal estuaries for striped bass occur from the Gulf of Mexico through the St. Lawrence Estuary. Exhibiting partial migration throughout their range [20,34,35], migratory (oceanic) striped bass are most abundant in the US Mid-Atlantic Bight [36], but are also common in the Gulf of Maine and Canadian waters [37,38]. The Hudson River, Chesapeake Bay, and Delaware Estuary together support the US Atlantic shelf stock, which is assessed and managed by the Atlantic States Fisheries Management Commission [36]. In 2019, overfishing was assessed for the stock, which currently supports the US's single most valuable marine recreational fishery [36]. Important commercial fisheries also occur, principally in the Chesapeake Bay. The Hudson River Estuary supports its own population, which exhibits partial migration [21,39], partitioning into

Upper Estuary (resident), Lower Estuary, and Oceanic (migratory) contingents that persist over years and life-times. Past research shows that partial migration shapes overall striped bass population responses to pollution, fishing, storms and management [20,40,41]. In this study, we target the Oceanic contingent through selection of larger individuals during spring [42].

The Hudson River Estuary—a fjord valley- is long (243 km), linear, and deep in comparison to other Mid-Atlantic Bight estuaries, lending itself well to biotelemetry coverage (Fig 1). Each spring, spawning fish transit through New York Harbor (river km 0) into the Peekskill-Storm King highlands region (river km 70–90), where upon salinity rapidly diminishes [43]. Snowmelt and freshets cause large variation in the salt front position during this period, which can extend nearly to New York Harbor in extremely wet years. Striped bass eggs, larvae, and juveniles have been well monitored in the Hudson River Estuary, the result of an energy utility's agreement to conduct Hudson River longitudinal surveys beginning in 1974 [27,33]. Key spawning reaches are indicated from ichthyoplankton collections, with modes of egg density at about 90 and 120 km (Fig 1). Decades of monitoring spawning adults confirm the traditional occurrence of spawners in the 80–100 km and 160–200 km reaches [44]. Egg densities are likely displaced downriver from these spawning centers owing to down-estuary advection of eggs. Proximity to the salt front retains eggs spawned at the lower reach [25].

## Capture and tagging

During April and May 2016, 100 Hudson River striped bass were captured within the two spawning reaches (Fig 1) and received coded acoustic transmitters (Lotek Wireless, Inc.; model MM-MR-16-50; 8 cm, 35 g, 2.5 year expected battery life; continuous 60s transmission delay at 69 kHz, 7s delay from April-June at 76 kHz). All fish were > 68 cm total length (TL) and assumed to be members of the migratory Oceanic contingent [21,40]. As the two reaches differ in extent and bathymetry, they required that sampling take place during non-overlapping times and with different gear. Within the lower reach, 50 fish were captured in shallow water through electroshocking and tagged April 20–26. As waters warmed, fish moved into deeper waters and became inaccessible via electrofishing after April 27. Another 50 fish were captured in deeper water in the upper reach with a 152 m haul seine deployed by boat and tagged May 5–19.

Surgeries in both regions were conducted onboard a small vessel in a portable electro-immobilization unit [45] following procedures under an approved protocol by the University of Maryland Center for Environmental Science IACUC (#F-CBL-16-05). Fish selected for tagging were immediately transferred to a flow-through tank or in-river live well. Holding time both pre- and post- surgery was adequate to both ensure healthy condition (<5 minutes each) and to minimize holding stress on the fish. In preparation for surgery, fish were inclined on a surgery sling with a tank containing ambient freshwater so that their head and gills were immersed and abdomen exposed for surgery. Anesthesia using electronarcosis was immediate (<30 seconds); introduced voltage was adjusted to induce anesthesia but generally ranged between 15 and 20 volts. Sex was determined either by expressed gametes or confirmed during surgery. Length and weight measures were taken, and then sterilized transmitters were implanted through a 1–2 cm incision slightly lateral to the linea alba and mid-distance between the vent and pectoral fins. Incisions were closed with a series of simple surgical knots and sterilized. Fish were held in pre- and post-surgery recovery pens.

## Biotelemetry

During the spring 2016–2018 study period, 22–62 telemetry receivers per year (Vemco VR2W ©) were deployed throughout the Hudson River Estuary between river km 43 and 245 (Fig 1).

The two study spawning reaches were well bracketed by both down- and up-estuary receivers. At deployment sites above river km 100, the detection range of receivers (600 m) allowed bank-to-bank coverage by single receivers. This was not the case for those deployed below river km 100 where the estuary broadens (Haverstraw Bay). Receivers were hung on buoys, where they logged detections during deployment periods April-October each year. Receivers were checked every 2–3 months and detection data downloaded. Outside the Hudson River Estuary, deployments of acoustic telemetry arrays occurred throughout the US NW Atlantic shelf waters, supporting evaluations of broad scale coastal migrations. The Atlantic Cooperative Telemetry Network [20] is an online portal that facilitated the return of transmitter detections from colleagues within the Network willing to share data. Consistent array deployments occurred 2016–2018 and supported depictions of seasonal shelf migrations, ordered by latitude. For depiction purposes receivers were combined for Maine (ME; estuary and shelf waters), Massachusetts (MA; principally Massachusetts and Cape Cod Bays), Long Island Sound (LIS), NY Bight (NYB; combined New York and New Jersey shelf waters), Delaware (DE; estuary and shelf waters), Maryland shelf (MD), Virginia shelf (VA), and Chesapeake Bay (CH; estuary).

## Spawning run contingents

The design of the biotelemetry study sought to sample equivalent numbers, sex ratio, and size range between the two hypothesized spawning reach contingents. In addition to being a spawning area, the lower reach represents an area of staging for up-estuary-spawning fish. Due to temperature constraints mentioned above, the lower reach was sampled first. Therefore, an equivalent number of spawners in each reach were targeted in year 2016, recognizing that without *a priori* information on how fish visited each area it was likely that lower reach sampling may have captured spawners destined for the upper reach. To correct for this bias, we classified spawners during the subsequent spring spawning run of 2017. Those individuals were then compared with their 2016 capture location and subsequent 2018 spawning run behaviors.

Before classifying 2017 spawning run contingents, the mean daily river kilometer of each fish's observed detections was calculated. Gaps in the daily time series were interpolated with a 4-day exponential-weighted moving average using the *imputeTS* package in R [46]. Per-fish, this resulted in 23.9 ± 9.9 and 26.0 ± 6.1 measured observations (mean ± standard deviation) from river entry to exit in 2017 and 2018, respectively, and 14.0 ± 7.8 and 8.4 ± 8.5 imputed values. The ends of each series were catenated to equal length with the river kilometer of the New Jersey-New York border (river km 35) to maintain information on date and penalize alignment of run phenologies that occurred far apart in time. This point was south of the lowest receiver (river km 40) near Tappan Zee Bridge (river km 43), the designated start and end point for each spawning phenology.

Individual spawning phenologies were categorized by clustering around median centroids (*k*-medoids [47]) utilizing dynamic time warping as a time series dissimilarity measure. Dynamic time warping, a machine learning algorithm most often applied in speech recognition classification, was chosen due to its suitability in matching phenomena that are offset in time and magnitude [48,49]. Starting from the beginning of the time series, dynamic time warping iteratively finds which points are matched with the least cost [50]. This allows matching of multiple-to-one or one-to-multiple points (Fig 2), and reports the cost of "warping" the two time series to be alike in this manner. Compared to Euclidean matching, which necessitates one-to-one matching in time and would calculate dissimilarity between spawning run contingents based on the daily river kilometer distance between each fish, dynamic time

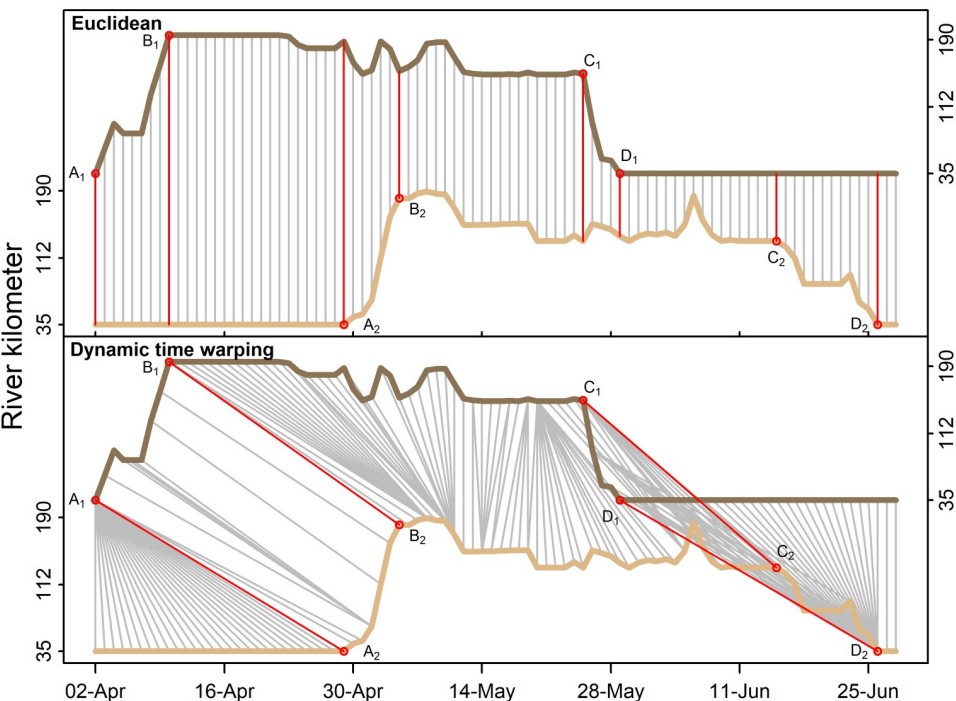

**Fig 2. Visualization of matching two spawning phenologies via standard Euclidian (top panel) and dynamic time warping (bottom panel) methods.** Single fish spawning phenologies are depicted as light brown (left axis units) and dark brown (right axis units). Time series are vertically offset for ease of visualization. Gray lines display daily positions matched using the respective algorithms. The arrival of each fish (subscript 1, 2) in the river (A) or spawning reach (B), and subsequent exits from the spawning ground (C) and river (D) are shown. Note that dynamic time warping allows these spawning phenologies, offset in time, to be matched, while the Euclidean method matches only those points that co-occur in time.

warping measures how much an entire spawning phenology trajectory must be stretched or compressed in time to match that of another fish. As such, spawning run phenologies that are alike in shape, extent, and destination, but offset in time are clustered together [51,52; Fig 2). The iterative clustering procedure, utilizing dynamic time warping and performed with the *dtw* [53] and *dtwclust* [54] packages in R, solved for maximum within-cluster similarity and minimum between-cluster similarity. A synthetic median centroid was extracted from each of two clusters to represent lower- and upper-reach spawning.

To investigate consistency of run contingent categorization for individual fish across years, the tracks of returning fish in 2018 were clustered onto the 2017 centroids and the cross-classification between the two years compared. Independence between cross-classified frequencies between years was tested using a Chi-square test.

## Spawning run phenology

Temperature was examined as a possible driver for the phenology of 2017-defined run contingents, under the premise that the 2017 classifications best represented the two run contingents. Flow was also examined for its influence on temperature. All classified individuals were included in 2017, and only those individuals correctly classified included for 2018. Daily mean water temperature (°C) and discharge ($m^3s^{-1}$) records encompassing the spawning season

were extracted from USGS Poughkeepsie water monitoring station (river km 122; Fig 1) and compared to the cumulative presence of fish in each run. Runs entered the spawning reaches in a series of pulses (see Results) not well characterized by weighted-means that are commonly used to describe spawning phenology [55,56]. Rather, time of entry and exit were characterized by median dates and experienced temperature by the 50th percentile occurrence for each run contingent.

### Mortality between spawning run contingents

Attrition of tagged fish was followed over the two years after release, with the intent focused on evaluating differential mortality between spawning run contingents following their identification in spring 2017. Mortality was analyzed for the period June 2017-December 2018 to limit any bias associated with the tagging procedure. The last detection date for each individual separated their period at large (alive) and their assumed loss from the sample (death). The number of extant individuals was summed for each date. Sums were log-transformed and regressed against days-at-large (date) to estimate daily instantaneous loss rates (Z). Post-spawn survival was modeled for the clusters identified in 2017 separately for each year using Kaplan-Meier estimation. Differences in survival between the clusters was tested using the Peto & Peto modification of the Gehan-Wilcoxon test. Survival analyses were conducted using the *survival* package in R [57].

## Results

### Biotelemetry returns

Tagged fish in the upper reach (n = 50; 71–104 cm, mean 88.6 cm TL; 3.5–17 kg, mean 8.6 kg) were significantly larger than those in the lower reach, although sizes broadly overlapped (n = 50; 68–99 cm, mean 83.4 cm TL; 2.8–12.1 kg, mean 7.0 kg) (Fig 3; Welch's t-test: p = 0.002, n = 100; p = 0.003, n = 97, respectively). Females were 11.0 cm and 3.2 kg larger than males at both reaches (Welch's t-test, combined reaches: p<0.001, n = 100; p<0.001, n = 97, respectively). In both reaches, more females were captured and tagged than males (lower reach: 28 female, 22 male; upper reach: 36 female, 14 male), and sex ratio did not vary between reaches (Chi-squared: p = 0.10, n = 100).

Telemetry returns in the two months following tagging indicated that all but one fish—which stopped being logged shortly after surgery—migrated downriver to regions below the Tappan Zee Bridge (river km 43); seven others were not detected after leaving the Hudson River that year. Sixty-six fish completed the spawning run the following spring (2017), while 40 fish completed it in 2018. One tagged fish from each region skipped spawning in 2017 (i.e., did not return to Hudson River spawning reaches), but did return during spring 2018. Four other fish, two tagged in each Hudson spawning region, either strayed or were originally strays from the Chesapeake and Delaware estuaries and did not return in subsequent years.

### Spawning run contingents

In 2017, time series clustering of returning tagged striped bass produced two distinct median behaviors (Fig 4). Centroid 1 was the most frequent behavior (45/66) and is clearly oriented towards the upper reach, with individual phenologies showing repeated up-estuary excursions but tending not to retreat to regions below river km 150. The median period of the first classified run contingent, as defined by centroid 1, is April 22 –May 30, with individual phenologies ranging between April 4 and June 24 (Table 1). Centroid 2 represented the minority of fish (21/66 individuals), exhibiting a modal behavior centered on the lower reach. Similar to

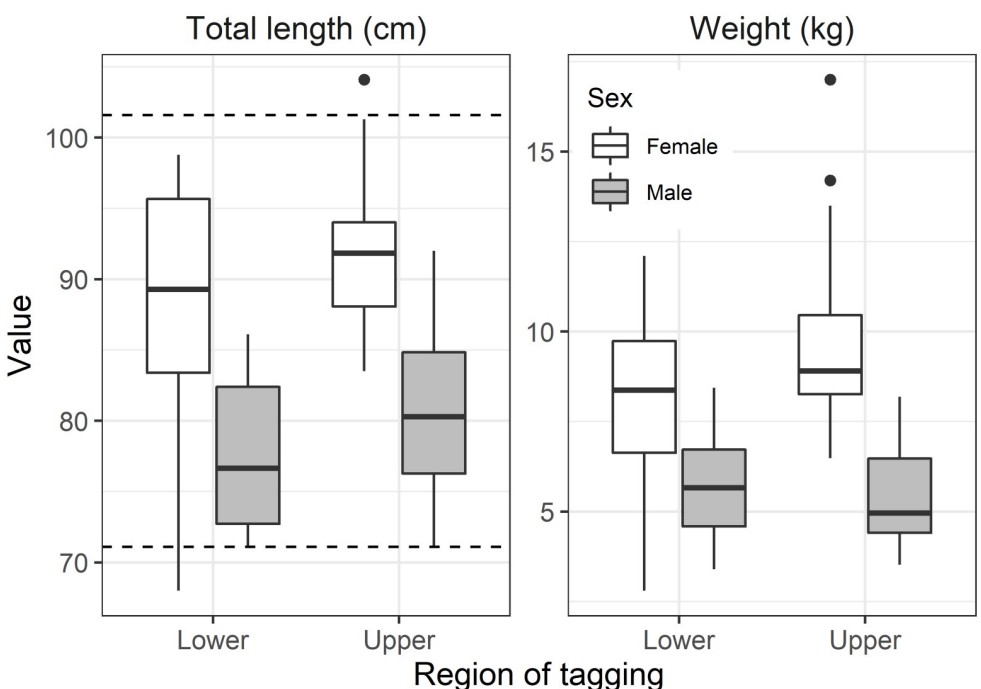

**Fig 3. Distributions (box whisker plots) of total length (cm; left panel) and weight (kg; right panel) of Hudson River striped bass receiving transmitters by tagging region and sex.** Horizontal dashed lines represent the Hudson River no-take slot limit between 71.1 and 101.6 cm Total Length.

Centroid 1, individual excursions occur up-estuary, with the retreat migrations delimited at c. river km 95. Centroid 2 represented an earlier run contingent behavior, with the median period April 18-May 21. Sex ratios were again skewed towards females, more so for the upper reach contingent (lower reach: 11 female; 10 male; upper reach: 29 female, 16 male), but did not signicantly differ (Chi-squared: p = 0.51; n = 66).

The 2017 and 2018 clustering procedures provided similar individual classications with all but 4 of the 38 individuals that returned in both years (two of the 40 fish returning in 2018 had not returned in 2017) correctly classified (11% misclassification; Chi-square test: P<0.001; Figs 4 and 5). All misclassified individuals in 2018 occurred for the lower reach contingent (4 of 12 individuals; Fig 5C). In 2018, upper reach fish were again more frequent (n = 26) in comparison to lower reach individuals (n = 12). Median periods were April 20-May 27 for lower reach and April 27-May 28 for the upper reach.

Centroids classified for 2017 and 2018 identified upper and lower reach fish, whose membership was tested against individuals tagged in upper and lower reach samples in 2016. Chi-square tests showed that tagging location influenced identified centroids for both 2017 and 2018 (P<0.01). Thus, tagging locations in 2016 were reasonably selective for upper and lower reach contingents (Fig 5). Most fish (89%) tagged in the upper reach in 2016 were classified (in 2017) as Centroid 1, while only 55% of fish tagged in the lower reach in 2016 were classified as Centroid 2 in keeping with the prediction that some fish sampled in the lower reach were intercepted enroute to upper reach spawning. In 2018, 78% (31/40) of classified 2018 fish aligned with 2016 capture and release locations, and were similarly better classified for upper reach (88%) than lower reach (63%) samples.

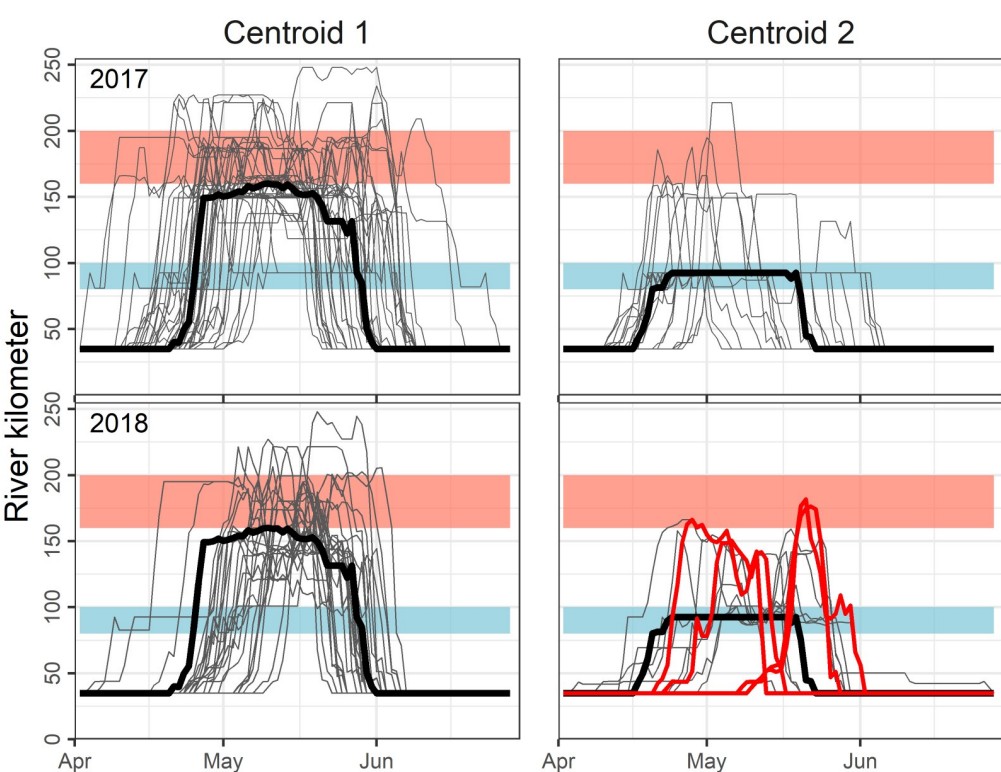

**Fig 4. Dynamic time warping centroid identification of Hudson River spawning run contingents.** Centroids are fitted to 2017 spawning run behaviors in both 2017 and 2018 (bold black lines; note centroid phenology is duplicated for both years). Individual phenologies are shown as gray and red lines, the latter indicating mis-classified individuals. Colored rectangles identify the river kilometers of the upper (red) and lower (blue) reaches.

## Spawning run phenology

Surface water temperature in both years increased from 4°C to 23°C as fish moved up-estuary during the spawning season (Fig 6). In 2017, temperature increased more rapidly during April than in 2018, approaching 13 °C by April 28, then leveling off during early May in association with increased discharge. In 2018, temperatures did not approach 13 °C until May 13 and showed steady increases throughout spring.

The upper reach contingent lagged behind the lower reach contingent, measured as 50% incidence, more so in 2018 than in 2017 (Fig 6). Different responses by contingents to temperature may be indicated for the upper reach contingent, which shifted its timing earlier during the warmer year (April 22, 2017) in comparison to the cooler year (April 27, 2018). In contrast, the lower reach contingent exhibited the same date of 50% incidence between years (April 18).

**Table 1. First, last, and median dates of entry above, and exit below, Tappan Zee Bridge (river km 43) for the 2017 and 2018 spawning seasons.**

| Year | Centroid | Reach | First Entry | 50% Incidence | Last Entry | First Exit | 50% Exit | Last Exit |
|------|----------|-------|-------------|---------------|------------|------------|----------|-----------|
| 2017 | 1 | Upper | April 4 | April 22 | May 3 | May 19 | May 30 | June 24 |
| | 2 | Lower | April 12 | April 18 | April 28 | May 2 | May 21 | June 4 |
| 2018 | 1 | Upper | April 5 | April 27 | May 11 | May 11 | May 28 | June 5 |
| | 2 | Lower | April 5 | April 18 | May 1 | May 18 | May 27 | June 26 |

**A** Tagged

|  | Lower | Upper |
|---|---|---|
| **2017 Cluster** Lower | 17 | 4 |
| **2017 Cluster** Upper | 14 | 31 |

**B** Tagged

|  | Lower | Upper |
|---|---|---|
| **2018 Cluster** Lower | 10 | 3 |
| **2018 Cluster** Upper | 6 | 21 |

**C** 2018 Cluster

|  | Lower | Upper |
|---|---|---|
| **2017 Cluster** Lower | 8 | 0 |
| **2017 Cluster** Upper | 4 | 26 |

**Fig 5. Cross-classification of regions at tagging and as determined by 2017 and 2018 clustering of spawning run contingents.** (A) 2017 spawning run centroids v. 2016 tagging locations; (B) 2018 spawning run centroids v. 2016 tagging locations; (C) 2017 spawning run centroids v. 2018 spawning run centroids.

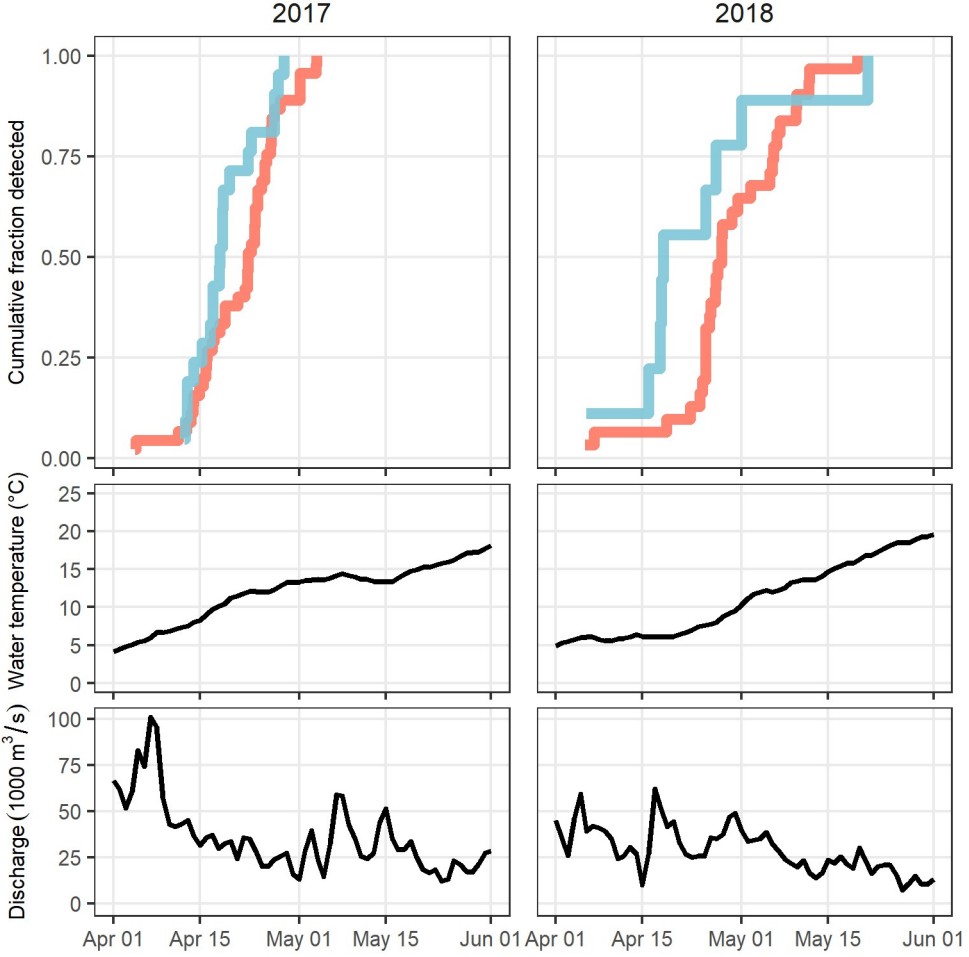

**Fig 6. Environmental conditions encountered in 2017 and 2018 by lower (blue) and upper (red) reach contingents.** Run contingents were classified according to 2017 returns. Top panels are cumulative frequency distributions on each contingent (lower: blue; upper: red), initiated as they pass Tappan Zee Bridge (river km 43). Monitored water temperature and discharge are daily means derived from Poughkeepsie USGS station data (river km 122).

With the exception of the lower reach contingent in 2017, a few individuals entered the river in early April when mid-river water temperatures were 5–6 ˚C. In 2017, mean temperatures at 50% incidence were 10.1 and 11.8 ˚C, respectively for the lower and upper reach contingents. In 2018, 50% incidence occurred at substantially cooler mean temperatures of 6.1 (lower reach contingent) and 8.0 ˚C (upper reach contingent).

## Coastal migrations

Detections in the three years following tagging (2016–2018) indicated that the two classified spawning run contingents undertook similar shelf migrations from southern Maine to southern Virginia (Fig 7). In 2016, we reclassified individuals to spawning run contingent on the

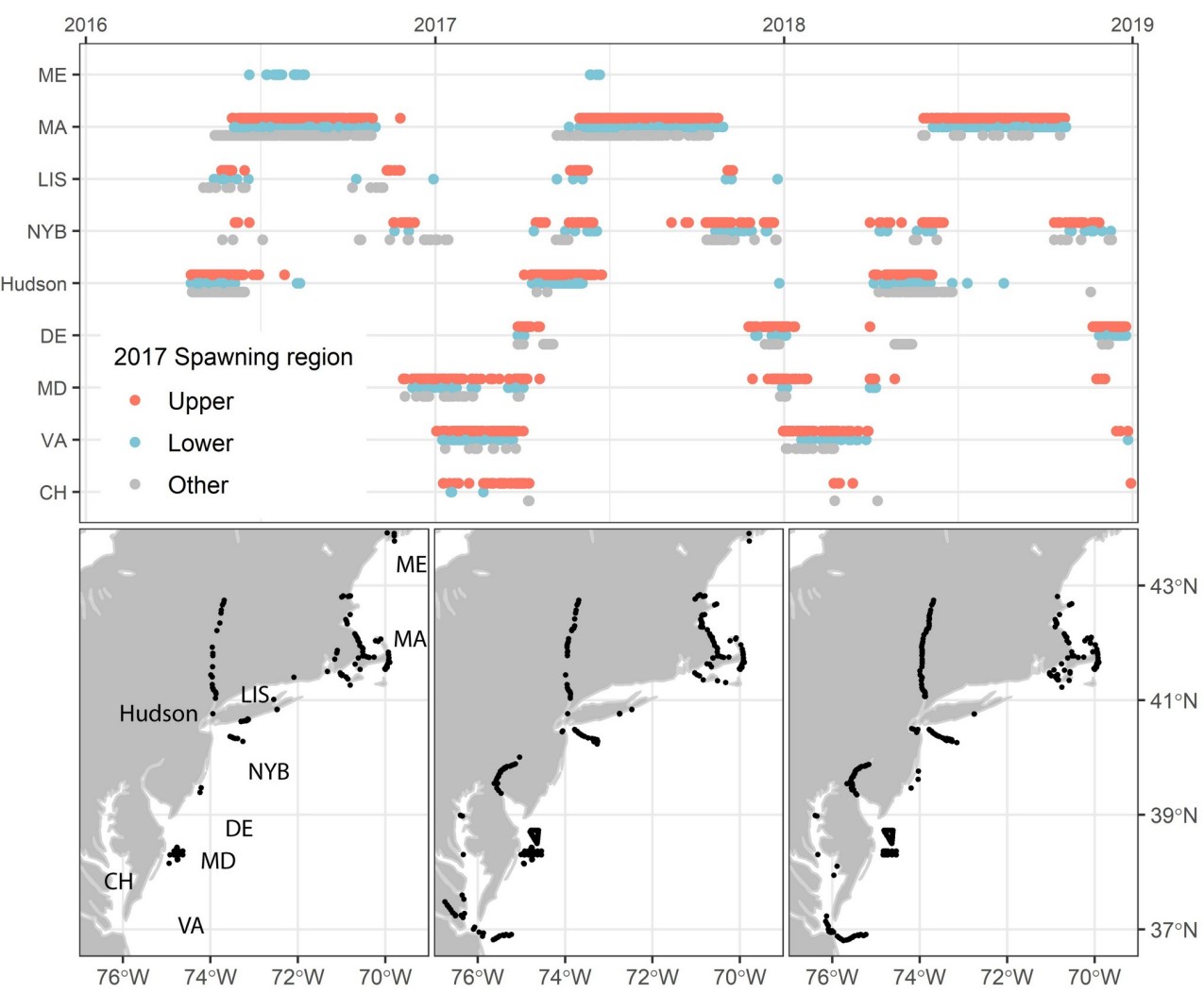

**Fig 7. Atlantic shelf water telemetry detections of Hudson River striped bass spawning reach contingents.** Top: Detections through time. Upper and Lower reach contingents are defined by clustering of 2017 spawning phenologies. The "Other" category are those not included in the 2017 classification owing to tag loss, skipped spawning, or straying. Detections are ordered by latitude of telemetry receiver array (Maine–Coastal Virginia/Chesapeake Bay). ME = Maine estuary and shelf waters, MA = Massachusetts and Cape Cod Bays, LIS = Long Island Sound, NYB (NY Bight) = New York and New Jersey shelf waters, Hudson = Hudson River Estuary; DE = Delaware River and shelf waters, MD = Maryland, VA = Virginia, CH = Chesapeake Bay. Bottom: Distribution of each year's detections with regional identification of receiver arrays (left bottom panel).

basis of the 2017 centroid analysis rather than tagging location and observed that the lower reach contingent departed the Hudson River estuary 6 days earlier than the upper reach contingent, similar to what was observed in 2017 when the lower reach contingent departed 9 days earlier on average. The opposite occurred in 2018, however, with the upper reach contingent departing 2 days earlier than the lower reach contingent on average. A single lower reach fish entered waters off Maine in the summers of 2016 and 2017, otherwise the pattern of shelf migrations in late summer, fall, winter, and early spring were quite similar between spawning reach contingents (Fig 7). Fish that left the Hudson River moved along southern Long Island into Massachusetts waters from May through June with an average transit rate of 10 km d$^{-1}$, where they remained until late September. After leaving coastal Massachusetts, tagged striped bass moved south at an average rate of 6 km d$^{-1}$, entering the coastal waters of Virginia off the mouth of the Chesapeake Bay in early-to-mid January. Striped bass remained in this area until mid-March or early April, when they rapidly moved back to the Hudson River at an average rate of 11 km d$^{-1}$. This pattern repeated each year with relatively little geographic or temporal variation.

## Mortality between spawning run contingents

The lower reach contingent experienced a high level of loss during the May-June period in 2017 and 2018 (Z = 4.9 10$^{-3}$ d$^{-1}$; Z = 7.4 10$^{-3}$ d$^{-1}$), suggesting periods of intense vulnerability (Fig 8). In comparison, the upper reach contingent experienced lower mortality during this period in 2017 (Z = 1.0 10$^{-3}$ d$^{-1}$), but similar losses in 2018 (Z = 4.5 10$^{-3}$ d$^{-1}$). Annual post-run Kaplan-Meier survival estimates differed between spawning run contingents only for 2017 (p = 0.02; 2018: p = 0.7). In 2017, annual post-run survival was estimated at 57.1 ± 10.8% and 82.2 ± 5.7%, for the lower and upper reach contingents, respectively. In 2018, annual post-run survival was similar between contingents (lower: 54.5 ± 15.0%, upper: 51.6 ± 9.0%). Note that the median exit date for the lower reach contingent was 9 days earlier than the upper reach contingent in 2017, and that the two contingents exhibited similar exit dates in 2018 (Table 1).

## Discussion

The three-year biotelemetry study uncovered discrete contingents within spawning runs, a behavior not yet documented for striped bass. Seasonal and sub-seasonal spawning runs are well described where they are overt, such as salmon streaming past the viewing window of a fish lift, or through seasonal fisheries that target spring, fall, or winter spawning aggregations of herring and cod. Here, biotelemetry coupled with the dynamic time warping cluster analysis exposed more-cryptic spawning run contingents that broadly overlapped within the same spring season, but used different reaches of the Hudson River estuary, and exhibited characteristic phenologies. We discovered that individuals largely participate in the same spawning run contingent year after year and that contingent membership had ecological consequences. In one of the two study years, the early run contingent was exposed to greater mortality than the later contingent -associated with early summer shelf fisheries. These differential demographic fates between run contingents can alter population outcomes, particularly if selective mortality occurs for one or the other contingent.

Mechanisms contributing to individual fidelity to either spawning reach contingent were not evaluated but could relate to learned behaviors and tradition. Given the level of infidelity between run contingents (11%), particularly for the lower reach contingent (25%), and high dispersal and mixing of embryos and larvae [25], persistence in contingent behavior is unlikely related to genetic lineage. For first maturing individuals, associative schooling behaviors with larger and older individuals within the Hudson River estuary could promote initial adoption

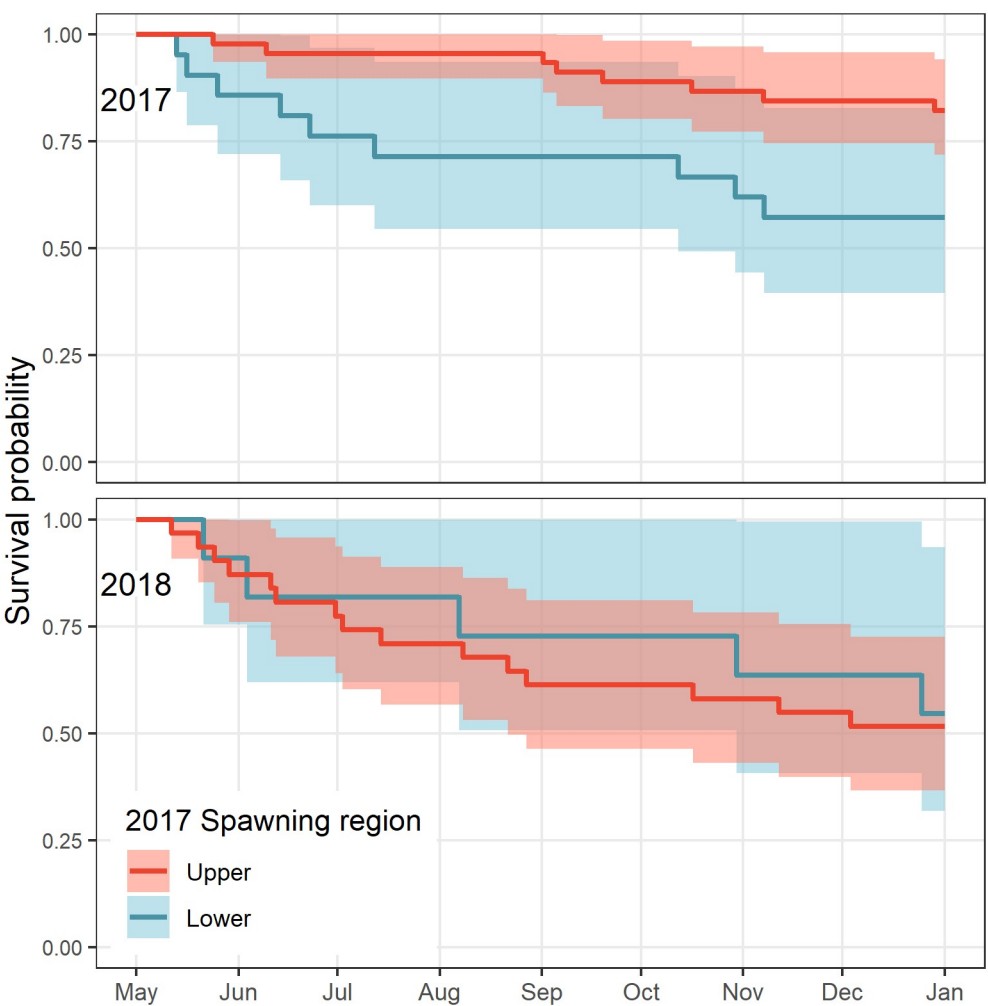

**Fig 8. Kaplan-Meier curves of post-run fish by 2017 spawning run contingents classification in 2017 and 2018.** Lines are coded red for the classified upper reach and blue for the lower reach contingents. 95% confidence intervals are shown as envelopes.

of either spawning run behavior [29,58,59]. Thereafter spawning reach fidelity may relate to key environmental differences in conductivity, tidal flow, bathymetry and channel dimensions that allow piloting and then navigation to particular spawning reach features. In past telemetry studies, striped bass have shown remarkable fidelity to specific locations summer after summer, indicating precise navigation is within their behavioral repertoire [21,42,60]. Such precise homing behaviors are of course well known for salmon species [61].

## Spawning run classifications

The identification of modal migration behaviors within spawning runs and individual fidelity across years required a more nuanced classification approach owing to the regional and temporal overlap between the contingents. We initially considered a more traditional time series hierarchical clustering analysis [62], but found that the approach was too sensitive to unequal phenology durations and small misalignments between daily positions (Fig 2). Dynamic time warping evaluates the degree with which time series must be warped or bent to make them

equivalent, and does not have the requirement of equal-length time series [48]. Developed as a machine learning tool in speech recognition applications, it has received scant attention by movement ecologists [51,52]. In its application to inverted U-shaped spawning run behavior, the algorithm performed well, summarizing time series that ranged between 16 and 70 days in duration, and time series that deviated from symmetry and often showed multiple up-estuary excursions (Figs 2 and 4). The approach captured and characterized the two hypothesized reach-specific behaviors despite high variance in individual movements.

Implementing the dynamic time warping approach required complete and overlapping time series for each fish entering the Hudson River. The high longitudinal coverage of the Hudson River Estuary by fixed telemetry receivers aided here, but times series catenation to a start and end point, the Tappan Zee Bridge (river km 40), and interpolation of missing daily records were still required of the approach. Our analysis was informed by the expectation of only two dominant behaviors, and a more refined cluster analysis (k>2), would likely support minority behaviors within these clusters for each year [21]. Still without expectation for additional spawning group behavior, we curtailed our comparison to upper and lower reach groups. This prediction was upheld by the membership persistence within identified groups.

## Environmental drivers

Striped bass spawning runs occurred during springtime warming, which ranged 4˚C to 23˚C across individual spawning phenologies. A slower rate of warming in 2018, was associated with a 5 day lag of the upper reach contingent, but the timing of the lower reach contingent was similar between years. This suggests that spawning run phenologies are influenced, but not tightly cued by temperature. In both years, a minority (<10%) entered early as temperatures ranged 4–8 ˚C. Entry during winter regime temperatures aligns with historical Hudson River fisheries which targeted fish under ice-cover [22]. Spawning migrations under ice cover have been noted for Canadian systems as well [37]. In both years, the number of fish participating in spawning runs increased rapidly between 10 and 12 ˚C. Then, most fish departed by the beginning of June as temperatures approached 20 ˚C. The lower reach contingent likely spawned at lower temperatures than the upper reach contingent with the difference larger in 2018. A recent historical analysis of spawning temperatures in the Hudson River Estuary recorded a range of 11.9 to 23.0 ˚C [33]. In more intensively studied Chesapeake Bay tributaries, spawning is associated with several-day surges in water temperature within this range [31,63]. In other literature, 12–20˚C bracket viable temperatures for embryo and larval growth and development in the Hudson River [32,64] and elsewhere [31,65–67] and in general the most favorable temperatures for larval survival occur between 15 and 19 ˚C [63], a period of maximum presence by spawning runs in both study years.

Despite general alignment between spawning runs and temperatures conducive for early survival and growth, the 6–9 day difference in contingent timing observed in 2017–2018 could result in differential outcomes on larval growth and mortality. Striped bass are capital spawners, with time of spawning sometimes mismatched to thermal or flow conditions favorable to larval survival [63]. Secor [68] suggested that a protracted spawning season maintained by a size-specific spawning phenology (larger fish spawning earlier [26]) could hedge against this mismatch. Further, warming in the Hudson River Estuary during the recent period (1976–2012) has resulted in a 7-d shift towards earlier spawning [33], increasing the opportunity for spawning-larval survival mismatches. Here, we observed differing spawning run phenologies that could mitigate against "mistimed" spawning. Because larval vital rates are quite sensitive to temperature, the 2–4 ˚C difference in temperatures observed between spawning reaches in

2017 could have resulted in an order of magnitude difference in larval and juvenile production [63,69,70].

## Spawning run contingents

Hudson River striped bass exhibited distinct spawning run contingents despite broadly overlapping coastal distributions. The term contingent applies here as population sub-components defined by persistent migration differences [71]. Contingent behaviors can shape overall population outcomes despite periods of high overlap in incidence. In 2017 the departing lower reach contingent entered shelf waters before the upper reach contingent and encountered recreational fishing that selected all sizes of the run (TL>71 cm). In contrast, the in-river fishery was restricted to fish both smaller, 41–71 cm TL, and larger >102 cm TL, protecting the bulk of spawners (Fig 3). We speculate that by departing the Hudson River and entering coastal fisheries 6–9 d earlier in 2016 and 2017, the lower reach contingent experienced greater fishing mortality than the upper reach contingent. Higher vulnerability would persist as this contingent successively entered New England state fisheries as it migrated northward. In 2018, when little difference occurred in median exit dates, mortality was similar between contingents suggesting similar exposure to these fisheries. Directed fishing in the river itself is largely focused in the upper reach [72], such that any latent mortality associated with catch and release of larger spawners would likely bias mortality opposite to observed pattern of contingent mortality. As further circumstantial evidence of higher selectivity on early departing striped bass, consider the "other" (undefined) category, which principally comprised individuals that did not survive until the following spring and therefore could not be included in the clustering procedure. In both 2016 and 2017, other category individuals departed to shelf fisheries at substantially earlier dates and may have been exposed to higher effort resulting in their loss (Fig 7). As Hudson River striped bass move into the mixed stock fishery, fishing mortality is predicted to predominate over "natural mortality" sources for > 7 year old cohorts [36].

During late summer, fall, winter, and spring—the two defined spawning run contingents showed similar north-south shelf migration patterns, matching those reported for migratory striped bass tagged in Southern New England shelf waters and the Potomac River [20,73]. These similar coastal distributions again suggest that differential survival in 2017 was likely related to distinct timing of Hudson River emigration.

The two spawning run contingents could serve to stabilize the overall Hudson River population owing to the portfolio effect. Here, contingents encounter varying mortality and reproductive success but jointly, their outcomes buffer the population under nonstationary mortality regimes. In instances where one contingent is substantially smaller in number, such as indicated here for the lower spawning run contingent, the portfolio effect is diminished, although resilience (persistence) is still enhanced [7,74]. Still, our sampling frame was curtailed to a single year's tagging effort and egg densities suggest a more equitable spread between spawning reach contingents during the past several decades (Fig 1). Early or later run contingents will interact with climate and anthropogenic forces with outcomes that vary year to year. Early survival and recruitment are sensitive to spawning phenology when thermal and flow conditions are highly variable year to year. We observed that early departure likely caused higher fishing mortality in the lower reach contingent in 2017, but in a scenario of future climate change and warmer springs [33], reproduction and early survival could be favored in the lower reach owing to reducing the current risk of spawning in cold sub-lethal temperatures [31,63]. Facilitating the portfolio effect by promoting contingent conservation is a novel construct in fisheries conservation [5,8,74]. The periodic aggregation behaviors by post-spawning striped bass likely exposes them to a period of intense exploitation, which could select and

diminish one or the other contingents. Size and season limitations could conserve "contingents for contingencies," [29] favoring population stability in the face of future and uncertain exploitation and environmental regimes.

## Acknowledgments

We are particularly grateful for numerous investigators who graciously shared data with us from receivers they deployed throughout the Mid-Atlantic and southern New England, primarily through the Atlantic Coastal Telemetry Network. They include Gail Wippelhauser (Maine Department of Marine Resources); Bill Hoffman, Sara Turner, and Greg Skomal (Massachusetts Division of Marine Fisheries); Jeff Kneebone (Anderson Cabot Center for Ocean Life); Megan Winton (Atlantic White Shark Conservancy); Micah Kieffer (USGS, Conte Anadromous Fish Research Laboratory); Evan Ingram, Robert Cerrato, Matthew Sclafani, Justin Bopp, Catherine Ziegler, Sara Cernadas-Martin, Keith Dunton and Mike Frisk (Stony Brook University and Monmouth Universities); Jake LaBelle (Wildlife Conservation Society); Danielle Haulsee and Matt Oliver (University of Delaware, supported through BOEM Project M16AC00009); Ian Park (Delaware Division of Fish and Wildlife); Rob Aguilar, Michelle Edwards, Kier Heggie, Mike Goodison, Kim Ritchie, and Matt Ogburn (Smithsonian Environmental Research Center); Carter Watterson (US Department of the Navy); Chris Hager (Chesapeake Scientific); Pat McGrath, Kevin Weng, and Dan Crear (Virginia Institute of Marine Science); Jonathan Colby (Verdant Power); and Justin Krebs (AKRF, Inc.). Kathy Hattala (past scientist, New York State Department of Environmental Conservation) provided key guidance and assisted in procuring support for this study. Cameron Freshwater and one other referee substantially improved the manuscript through their insightful and constructive comments.

## Author Contributions

**Conceptualization:** David H. Secor, Michael H. P. O'Brien, Amanda L. Higgs.

**Data curation:** Michael H. P. O'Brien, Benjamin I. Gahagan, Dewayne A. Fox, Amanda L. Higgs, Jessica E. Best.

**Formal analysis:** David H. Secor, Michael H. P. O'Brien.

**Funding acquisition:** David H. Secor, Amanda L. Higgs.

**Investigation:** David H. Secor, Jessica E. Best.

**Methodology:** David H. Secor, Michael H. P. O'Brien, Amanda L. Higgs, Jessica E. Best.

**Project administration:** David H. Secor, Amanda L. Higgs, Jessica E. Best.

**Resources:** David H. Secor, Benjamin I. Gahagan, Dewayne A. Fox, Jessica E. Best.

**Software:** Michael H. P. O'Brien.

**Validation:** Michael H. P. O'Brien.

**Visualization:** Michael H. P. O'Brien.

**Writing – original draft:** David H. Secor, Michael H. P. O'Brien.

**Writing – review & editing:** Michael H. P. O'Brien, Benjamin I. Gahagan, Dewayne A. Fox, Amanda L. Higgs, Jessica E. Best.

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
