## [Decision Letter · Decision Letter 0]

2 Sep 2020

PONE-D-20-18274

Multiple spawning run behavior and population consequences in migratory striped bass Morone saxatilis

PLOS ONE

Dear Dr. Secor,

Thank you for submitting your manuscript to PLOS ONE. After careful consideration, we feel that it has merit but does not fully meet PLOS ONE’s publication criteria as it currently stands. Therefore, we invite you to submit a revised version of the manuscript that addresses the points raised during the review process.

Dear Dave,

We now have two reviews of your manuscript. Both the reviewers and I see potential for a good paper that should be published in PLOS ONE. However, the reviewers have some concerns and criticisms (see below). These need to be addressed in detail befor we can accept the ms. As far as I can see the reviewers' requirements do not demand new analysis, but as ref 1 writes: "My principal criticisms rest on the presentation and interpretation of particular aspects of the study". 

Best regards

Geir

We look forward to receiving your revised manuscript.

Kind regards,

Geir Ottersen

Academic Editor

PLOS ONE

Journal Requirements:

Reviewers' comments:

Reviewer's Responses to Questions

**Comments to the Author**

1. Is the manuscript technically sound, and do the data support the conclusions?

Reviewer #1: Partly

Reviewer #2: Partly

2. Has the statistical analysis been performed appropriately and rigorously? 

Reviewer #1: Yes

Reviewer #2: Yes

3. Have the authors made all data underlying the findings in their manuscript fully available?

Reviewer #1: Yes

Reviewer #2: Yes

4. Is the manuscript presented in an intelligible fashion and written in standard English?

Reviewer #1: Yes

Reviewer #2: Yes

5. Review Comments to the Author

Reviewer #1: Review of PONE-D-20-18294: Secor et al. 2020 – Multiple spawning run behaviour and population consequences in migratory striped bass

The submitted manuscript presents evidence of distinct spawning runs of Hudson River Estuary striped bass. The authors use two years of biotelemetry data and a dynamic time warping (DTW) analysis to identify distinct migration behaviours associated with population contingents that spawn in the upper and lower portions of the estuary. Evidence for differences in spatial distribution occurred in both years, while temporal differences were more ambiguous (the lower run migrated earlier in 2017, but not in 2018). Additionally, the authors attempt to link differences in phenology with abiotic environmental drivers and present evidence of differential survival among contingents despite similar coastal distributions. Overall, I found the manuscript to be informative, well written (particularly the introduction), and timely given continued interest in the potential for life history diversity to stabilize exploited populations via portfolio effects. I believe the analytical techniques are appropriate given the research questions and commend the authors on exploring novel modeling approaches that have the potential to be relevant to the broader ecological literature. My principal criticisms rest on the presentation and interpretation of particular aspects of the study. If these can be addressed, I believe the manuscript would be suitable for publication. Below I describe my major criticisms, followed by a summary of minor issues.

Major

1) Spawning runs vs contingents: Throughout the manuscript (e.g. line 24, 30, 61-63), the authors move back and forth between using the terms spawning run, spawning run contingent, and contingent to describe the different migration phenologies they observed. Given the first author’s original description of contingents in the 1999 paper, I believe that term is most appropriate, but do not believe it should be used synonymously with spawning run. At least within Pacific salmon, spawning runs are commonly used to describe genetically isolated groups, indeed run timing is one dimension used to define ESUs in the US and CUs in Canada, while the authors note that the upper and lower reaches are likely to have substantial genetic exchange. That being said I recognize that the definition of life history terms can vary region to region and taxa to taxa. Thus, my main recommendation would be to clearly define early in the introduction how each term relates to the other. Note that this may impact the manuscript’s title.

Also in the introduction, it would be beneficial to clarify how spawning location, phenology, or both contribute to classifying contingents in this context since temporal or spatial segregation could be used to classify a run, but evidence for each is somewhat ambiguous here (e.g. straying between groups, downstream advection of eggs, similar migration phenologies in one year).

2) Dynamic time warping – As the authors note, DTW is not commonly encountered in ecological analyses. Indeed, I had to spend a decent amount of time reviewing the relevant literature to understand what was happening under the hood. The authors briefly introduce the technique in the methods and provide some additional information in the discussion, but I believe an additional short paragraph summarizing the primary references would be beneficial for readers who are curious about the technique.

On a related note, I initially struggled to interpret Figure 2, but if adapted, I believe it offers an opportunity to introduce and clarify DTW to readers. First, I would recommend showing the Euclidean plot in the top panel since readers are more likely to be familiar with that distance metric. Second, it would be beneficial to add text insets (e.g., a, b, and c at the beginning, middle, and end of the time series), which are referenced and interpreted in the caption since readers are not fully familiar with the study’s data until the end of the methods, but Fig. 2 is referenced partway through. Finally, I would suggest summarizing what the implications of the Euclidean vs. DTW distance measures are in this simple example (either in the caption or when Fig. 2 is referenced in the text). For completeness, it may be helpful to add a second panel, which shows two time series where Euclidean and DTW estimates provide relatively similar conclusions to contrast visually with the example where they clearly are different. As an aside, in my version the second figure does not have any red or blue, just two shades of brown (I’m using a Mac so perhaps this is a PDF conversion issue?).

3) Environmental linkages – I understand the authors interest in identifying drivers of interannual variability in run timing and agree that discharge and temperature are two plausible candidates for consideration. However, I was surprised that the authors describe strong linkages between temperature and phenology in the discussion given their results. First, since the observational unit in this part of the analysis is an entire run (sampling population level response to a shared driver), it is difficult to draw robust conclusions about the relationship between drivers and phenology with only two years of data. Second, only the upper run delayed in 2018 and the 50% migration point was remarkably stable between years despite a 3-4 C difference in temperature (depending on the run). There was also considerable variation within a run, which suggests that even if temperature does cue the run, it likely only explains a relatively small component of the total variability. In other words, evidence of temperature cues seems to be weak and ambiguous and I actually was under the impression that they would discuss the stability of run timing despite interannual differences in temperature

Similarly, it seemed odd to me that temperature at the mid-point of the run was used as a proxy for thermal conditions that may cue both river entry and egress. Since the data suggest considerable variation in the rate at which warming occurs, it seems as though temperatures at the beginning and end of the summer should be correlated with entry and egress, respectively. Finally, I didn’t understand why discharge data were presented in Fig. 6, but not incorporated into the analysis since they appear to show substantial variation between years as well. If I am misinterpreting the authors analyses (which is possible) I would recommend they clarify their hypotheses and methods, and perhaps introduce additional language that emphasizes the limitations of their approach given the data available.

4) Differential mortality – I found the survival data very interesting and agree with the authors that it is consistent with differential mortality among the contingents, which is clearly of management interest. That being said, it’s not clear to me how a moderately earlier ocean entry (~seven days) led to sustained higher rates of mortality over ~two months for the lower reach group. If differential mortality was driven solely by built up recreational harvest then why wouldn’t the difference in survival be restricted to May/June, rather than continue through July (based on Fig 7)? Is it possible these are delayed mortalities from catch-release fisheries, particularly if lower estuary fish are exposed to higher in river harvest effort? Otherwise there must be some difference in exposure to fisheries, or natural mortality mechanisms, through the summer to drive a consistently divergent pattern; however, that doesn’t seem to be the case based on the coastal detections that are presented. More generally, I would caution the authors against emphasizing harvest as the driver of differential mortality unless they can present data distinguishing natural mortality from fisheries mortality within the tag groups (e.g. a larger proportion of lower reach tags returned by harvesters) or it is established that natural mortality for these age classes is truly negligible. If the latter, appropriate references should be added.

Minor

Line 41: Replace with “enter” singular.

Line 43: Suggest replacing “contributing to” to “potentially stabilizing” given the focus on hypothetical portfolio effects.

Lines 71-73: The phrase “centers” implies to me a multimodal distribution of spawning density; however, the egg data in Fig. 1 appear approximately normally distributed suggesting a continuum of spawning habitat rather than distinct spawning areas. That being said, the rationale in lines 116-118 seems reasonable, I would suggest adding a brief version to the figure caption to help confused readers.

Line 79-80: Suggest re-phrasing to “Points show yearly values of egg density and box plots represent interannual variation”.

Line 88-89: I find the phrase “characteristic migration modes” a little vague. Perhaps specify whether the interest is primarily variation in spatial distribution, temporal distribution, or both? Lines 71-73 suggest the interest is primarily spatial, but the first paragraph, and the bulk of the manuscript, include reference to timing as well.

Line 104: Strictly speaking where do lower estuary individuals fall on the resident/migratory continuum? Also 126 indicates all individuals were migratory ocean contingent. Since fish were tagged in the upper estuary, where individuals are defined as resident, how was this determined?

Lines 128-130: Any concerns of differential effects of seining vs. electrofishing?

Line 142: Duration of holding?

Line 146: Adding receivers to Fig 1 would be helpful.

Lines 152-161: I know that acquiring receiver metadata from tracking networks can be difficult, but if possible, a map showing the distribution of the marine receivers (similar to Secor et al. 2020) would be helpful.

Line 175: It might be useful to insert summary statistics on the extent to which individual time series had to be imputed (i.e. mean + SD of number of imputed days vs. observed).

Lines 183-185: As noted above, given the novelty of dynamic time warping in ecological studies (e.g. line 384) I think a paragraph or so explaining DTW (assumptions and basically what’s going on under the hood) is warranted.

Line 199: The language here is a little unclear to me because the authors do attempt to identify environmental drivers in 2018. I think some text is needed to clarify that the 2017 centroids were assumed to be correct. If this is the case though, it’s not clear to me how individuals that strayed (i.e. identified to one reach in 2017 and the other in 2018) were accounted for in this analysis. Additional clarification on this would be helpful.

Lines 248: I find the use of the term “individual spawning runs” a bit misleading because I most generally think of runs as describing behavior of multiple fish. Hence an individual run suggests, to me, a single categorical classification within a group (e.g. fall rather than spring or summer run), rather than an individual’s behavior. I would suggest replacing with individual migration phenologies, individual migrations, or a similar term as appropriate. Similarly, I think it may be more appropriate to phrase the following sentence as: “The median period of the first run, as defined by centroid 1, is…”

Line 261: Any reason not to include river km, instead of latitude, as the y-axis? It seems to be more commonly referenced in the text than latitude (e.g. line 297), is a bit more intuitive to readers, and presumably it would be an easy swap given the data necessary to make Figure 1. Also spawning is mis-spelled.

Lines 268-269: To clarify, the median periods described here are defined by the median centroids identified for 2018; however these are not shown in in Figure 4 or anywhere else, correct? Personally, I think it might be better to show the 2018 centroids in the lower panels of Figure 4, since these seem to be of more interest than the misclassification patterns (which can be easily described in the text). If this change is made then I believe Table 1 could be removed.

Line 298: Again, it may be nitpicky but I would recommend defining run as a phenotype or suite of migration behaviors, rather than an observation. I.e. the authors studied two runs in two years, rather than four runs. As the authors note in their introduction, interannual stability is a defining characteristic of migratory contingents so it seems inappropriate to consider these as independent replicates.

Lines 315: The phrase 2017-categorized is a little confusing initially. In the case of 2016 individuals, I take it to mean fish that were tagged in the lower portion of the river and then had a centroid 2 type behavior?

Line 359: The authors note that the midpoint of both runs in 2017 occurred at a similar temperature (1.7C difference). In 2018 it was a 1.8C difference. So I’m not sure the authors show strong evidence of a differential response to temperature given that interannual variability in temperature seems to overwhelm differences in the temperatures each run experienced.

Lines 360: Use of persist in this context seems a little out of place.

Lines 361-363: The authors don’t present any data that allows for natural mortality to be distinguished from fishing mortality (e.g. tags returned by harvesters). I would recommend either removing the emphasis on harvest or that the authors add citations demonstrating that the majority of adult mortality is associated with harvest. If the latter, I would still suggest adding qualifying language given our generally poor understanding of natural mortality rates.

Lines 368-369: I’m not sure I follow. I assume the authors mean that first-spawners adopt the behaviors of older individuals that are proximate when migrations begin. However, the two runs do not appear to be segregated spatially or temporally during coastal residence and upstream migrants necessarily bypass downstream spawning habitats so it’s unclear to me where proximity and abundance come into play. I recommend the authors refine this hypothesis a bit more.

Line 370: Strictly speaking, this doesn’t seem to be an alternative hypothesis, but rather an explanation for how distinct behaviors are maintained (while the former seems to be more focused on how they arise).

Line 373: Insert “behavioral” (or synonym) before repertoire.

Line 390: What’s meant by comparable here?

Lines 429-430: Again nitpicky, but it seems that by definition spawning runs will exhibit differential use (temporal or spatial) of a habitat. Suggest rephrasing to something like: “Here we show that Hudson River striped bass exhibit evidence of distinct spawning runs despite broadly overlapping coastal distributions”.

Line 436: I believe this should be Fig 3, however I also thought that Fig. 3 only provided an example comparison of two individuals. If so, then it seems like it would be better to reference Figure 4 and Table 1 (if retained). More generally I find the wording of this sentence confusing. It seems to imply that the lower run in 2017 is the only group to encounter the coastal fishery, which is clearly not the case. I would recommend rephrasing and moving the component on in-river vs. coastal slot limits to a separate sentence.

Line 436: Outmigration data from 2016 are not clearly presented in the results, except in Figure 7 which is difficult to interpret at fine temporal scales. I’d recommend adding somewhere if they’re going to be discussed in detail.

Line 444-447: I’d finish this paragraph off by clearly stating that the similar coastal distributions suggest that differential survival in 2017 is likely driven by early ocean entry phenology

Figure 7: I would recommend increasing the resolution on the final figure version and removing the panel borders that separate different years of observation. I think it would be better to show this as a one-panel figure with different years denoted along the x-axis so it’s immediately clear to readers that these are year-round observations (rather than 3 distinct tag deployments).

Table 1: I found Table 1 a little time consuming to interpret. I would recommend replacing with a figure providing equivalent summary statistics for easier interpretation. Alternatively, Figure 4 could be expanded to clearly show centroids in 2018, which would serve a similar purpose.

Reviewer #2: Summary

In their manuscript, Secor et al. describe their identification of two spawning subgroups within the Hudson River Estuary population of ocean-migrating striped bass that spawn annually. They suggest that these two subgroups are spatially segregated during spawning and follow similar migratory paths but exhibit different phenologies (i.e., staggered within the spawning season) as they enter and leave their respective spawning grounds. The mean daily latitudes of fish within these spawning subgroups were tracked with acoustic telemetry. Movement patterns of subgroups into the spawning reaches were then defined (upper and lower spawning runs) and time series differences analyzed using dynamic time warping. Individuals are proposed to maintain membership within established groups over time, based on 2 years of recapture data (2017-2018), where characterization of the upper versus lower spawning run subgroup was based on the 2017 spawning run. The lower spawning run occurred earlier (entered the study area and left the study area earlier) than the upper spawning run in 2017, but not 2018. They also find that these two subgroups follow similar routes along the Atlantic shelf outside of the spawning season. Moreover, the authors argue that different mortality rates between the subgroups help to stabilize the larger population dynamics.

Major comments

I think this is an interesting study that characterizes patterns of movement behavior for striped bass within the Hudson River Estuary and on the coastal shelf, and provides some support for spatial segregation of an upper and lower run during spawning. The manuscript was written well, and generally easy to understand. Survival analyses and conclusions from the survival analyses seemed appropriate (assuming that lower run fish were adequately detected, which was not clear to me based on Lines 149-150). The authors did a good job relating the increased mortality rate of the lower spawning group with the earlier (relative to the upper spawning group) entrance into shelf fisheries in 2017. This earlier migration behavior was potentially initiated in response to a steady increase in water temperatures throughout April that also triggered their earlier movement up-estuary to spawn.

A main concern is that the membership of individual tagged fish within the lower spawning run centroid was only consistent across 2017-2018 for 8 of 12 fish. Thus, although it is true that only 11% of individuals were misclassified between the two years overall, all of those misclassified were originally thought to be part of the lower spawning group, with few fish in that group overall. If the study was able to recapture fish in 2019, and there was evidence of continued site fidelity by those lower spawning group individuals, the case for spatially segregated groups with consistent membership would have been much stronger.

In addition, this group seems to have much lower membership overall, so I wonder how much it would be able to contribute to an overall stabilizing portfolio effect. If there is evidence of at times higher spawner abundances in the lower reach than in the upper reach (e.g., based on the decades of monitoring spawning adults mentioned in Lines 118-120), please include in Discussion. It is mentioned in Lines 452-454 that future climate change and warmer springs could lead to favoring the lower run. Please expand on how warmer springs could favor the lower run because it is not clear to me. Addressing these issues would help support the authors proposal that the two spawning runs could contribute to a portolio effect that stabilizes population dynamics for this contingent of striped bass.

Minor comments

• Line 116-117: “Spawning adults associated with these reaches are believed to be centered” – what is this belief based on? Observations from the authors?

• Line 117: Should 160-120 be 160-200? However, the lower boundary of the red rectangle in Figure 1 looks like it is greater than 160.

• Line 189: I am not seeing blue and red time series in Figure 2, as described here in the figure caption, just shades of brown.

• Line 213: Can leave out “excluding May”

• Line 293: I see a leveling off in Fig. 6 that starts the beginning of May and extends through mid-May, rather than a plateau only during mid-May

• Line 383-384: The authors included this statement: “Developed as a machine learning tool in speech recognition applications, we believe this is its [dynamic time warping] first application in comparing animal movement patterns.” As I wasn’t familiar with dynamic time warping, I did a quick search to see that it has recently been a suggested approach for analysis of animal movement as demonstrated within Cleasby et al. (2019), and applied to seabirds. I realize this is a recent paper, so understandably missed. I suggest the authors include this reference within the manuscript and adjust their statement accordingly.

References:

Cleasby et al. 2019. Using time-series similarity measures to compare animal movement trajectories in ecology. Behavioral Ecology and Sociobiology 73.

6. PLOS authors have the option to publish the peer review history of their article (what does this mean?). If published, this will include your full peer review and any attached files.

Reviewer #1: No

Reviewer #2: No

---

## [Author Response · Author response to Decision Letter 0]

2 Oct 2020

Please see attached file Response to Reviewers_Cot2

---

## [Decision Letter · Decision Letter 1]

26 Oct 2020

PONE-D-20-18274R1

Multiple spawning run contingents and population consequences in migratory striped bass Morone saxatilis

PLOS ONE

Dear Dr. Secor,

Thank you for submitting your revised manuscript to PLOS ONE. 

I have had one of the original referees review the new version of your manuscript. The reviewer is generally very pleased, but has a few minor points (s)he wants you to address.

When you have taken care of these I will recommend your manuscript for publication. I see no need for further review given that the remaining issues are minor.

We look forward to receiving your revised manuscript.

Kind regards,

Geir Ottersen

Academic Editor

PLOS ONE

Reviewers' comments:

Reviewer's Responses to Questions

**Comments to the Author**

1. If the authors have adequately addressed your comments raised in a previous round of review and you feel that this manuscript is now acceptable for publication, you may indicate that here to bypass the “Comments to the Author” section, enter your conflict of interest statement in the “Confidential to Editor” section, and submit your "Accept" recommendation.

Reviewer #1: (No Response)

2. Is the manuscript technically sound, and do the data support the conclusions?

Reviewer #1: Yes

3. Has the statistical analysis been performed appropriately and rigorously? 

Reviewer #1: Yes

4. Have the authors made all data underlying the findings in their manuscript fully available?

Reviewer #1: Yes

5. Is the manuscript presented in an intelligible fashion and written in standard English?

Reviewer #1: Yes

6. Review Comments to the Author

Reviewer #1: Again, I greatly enjoyed the opportunity to read this manuscript and commend the authors on their work. They thoroughly addressed my original comments and I particularly appreciated the additional clarity regarding DTW. I only have a few additional suggestions listed below.

Line 33-35: Some awkward wording in this sentence.

Line 354: Contingents mis-spelled as continents.

Line 357-359: Check edits.

Figure 7. The figure has been updated to include receiver detections in the authors’ reply (bottom), but not in the attached figure preceding the revised manuscript. Editorial office should double-check before proofing.

Line 431: Suggest inserting a but before used to contrast the contingents differences from their similarities.

Line 432-438: Suggest rephrasing to clarify that in one of two years the contingents were exposed to early summer fisheries at different rates, which was associated with greater mortality.

Line 531: Add “with” before catch and release.

7. PLOS authors have the option to publish the peer review history of their article (what does this mean?). If published, this will include your full peer review and any attached files.

Reviewer #1: **Yes: **Cameron Freshwater

---

## [Author Response · Author response to Decision Letter 1]

28 Oct 2020

All changes have been made according the reviewers helpful comments and edits.

Reviewer #1: Again, I greatly enjoyed the opportunity to read this manuscript and commend the authors on their work. They thoroughly addressed my original comments and I particularly appreciated the additional clarity regarding DTW. I only have a few additional suggestions listed below.

Line 33-35: Some awkward wording in this sentence.

Line 354: Contingents mis-spelled as continents.

Line 357-359: Check edits.

Figure 7. The figure has been updated to include receiver detections in the authors’ reply (bottom), but not in the attached figure preceding the revised manuscript. Editorial office should double-check before proofing.

Line 431: Suggest inserting a but before used to contrast the contingents differences from their similarities.

Line 432-438: Suggest rephrasing to clarify that in one of two years the contingents were exposed to early summer fisheries at different rates, which was associated with greater mortality.

Line 531: Add “with” before catch and release.

---

## [Editor Report · Decision Letter 2]

10 Nov 2020

Multiple spawning run contingents and population consequences in migratory striped bass Morone saxatilis

PONE-D-20-18274R2

Dear Dr. Secor,

We’re pleased to inform you that your manuscript has been judged scientifically suitable for publication and will be formally accepted for publication once it meets all outstanding technical requirements.

Kind regards,

Geir Ottersen

Academic Editor

PLOS ONE

---

## [Editor Report · Acceptance letter]

13 Nov 2020

PONE-D-20-18274R2 

Multiple spawning run contingents and population consequences in migratory striped bass *Morone saxatilis*

Dear Dr. Secor:

I'm pleased to inform you that your manuscript has been deemed suitable for publication in PLOS ONE. Congratulations! Your manuscript is now with our production department. 

Kind regards, 

on behalf of

Dr. Geir Ottersen 

Academic Editor

PLOS ONE